# An Ex-Situ Immobilization Experiment with Zn, Pb, and Cu in Dredged Marine Sediments from Bohai Bay, China

**Wensi Zhang, Xiao Wang and Baolin Liu \***

School of Marine Sciences, China University of Geosciences (Beijing), Beijing 100083, China;
zwensi@cugb.edu.cn (W.Z.); wangxiao2018@cugb.edu.cn (X.W.)

\* Correspondence: liubaolin@cugb.edu.cn; Tel.: +86-10-8232-2162

**Abstract:** The remediation of dredged marine sediments contaminated by metals has drawn increasing attention globally. Immobilization was regarded as a promising method for reducing adverse impacts on marine ecosystems. In this study, kaolinite and limestone were used as amendments to immobilize Zn, Pb, and Cu in dredged marine sediments, which were collected from the coastal zone adjacent to Tianjin Port in Bohai Bay. The sequential extraction procedure was applied to identify the mobility of metals and, further, to evaluate the immobilization effect of the amendments. The physical–chemical properties of the sediments, such as the pH, electrical conductivity (EC), salinity, and total organic carbon (TOC), were also measured to better understand their influence on the three metals' mobility. The results of the sequential extraction procedure indicated that the mobile fractions of the metals were converted into relatively stable fractions because of the two amendments. In addition, the EC, salinity, and TOC decreased moderately, while no obvious variations in the pH of the sediments were observed with the addition of kaolinite and limestone. It was confirmed that both kaolinite and limestone can effectively reduce the mobility and bioavailability of metals, particularly Zn, and limestone generally has a better immobilization effect, compared with kaolinite.

**Keywords:** dredged marine sediment; metal; immobilization; kaolinite; limestone

## 1. Introduction

A large number of dredged marine sediments are generated worldwide due to the management of channel navigation and harbor construction [1,2]. One of the main concerns raised in relation to dredged marine sediments is their potential risk to marine environments and organisms [3]. As a critical part of the geochemical cycle in marine ecosystems, marine sediment plays a role as the largest sink of both organic and inorganic pollutants [4,5]. Metals represent a crucial category of these pollutants, owing to their potential toxicity, durability, and bioaccumulation [6,7]. Once the sedimentary environment (e.g., pH, redox potential and dissolved oxygen) changes, metals in sediments can be released into the overlying water again and even enter the food chain, causing secondary pollution to surrounding areas [8,9]. Therefore, it is imperative to remediate dredged marine sediments contaminated by metals prior to final disposal [10].

A variety of technologies, including physical/mechanical [11], chemical [12,13], and biological technologies (phytoremediation and microbial remediation) [14,15], have been applied in the remediation of dredged marine sediments contaminated by metals. Among the various chemical remediation technologies, immobilization is considered a promising and alternative option due to its significant effect, low cost, and environmental friendliness [16]. The method involves the addition of chemicals to convert the mobile and bioavailable fractions of metals into stable fractions [17].

Successive chemical extraction techniques, such as the four-step BCR, on the basis of original three-step Community Bureau of Reference (BCR) [18], and the five-step extraction procedures, proposed by Tessier [19], have been widely used to assess the mobility and bioavailability of metals in marine sediments [20].

Immobilizing amendments decrease trace metals' mobility by several processes, which include adsorption, coprecipitation, ion exchange, and complexation reactions [21–23]. The most extensively applied amendments in marine sediments include iron-bearing materials, aluminosilicates, phosphates, and clay minerals [17,24]. For example, cement [25], lime [26], red mud, and apatite composite [27] were used for the remediation of polluted coastal sediments in Romania, Norway, and the Republic of Korea, respectively. While various kinds of amendments are being used, clay minerals have been extensively utilized for the immobilization of metals due to their simplicity of use, significant effect, and universal applicability [28]. As one clay mineral, kaolinite is used as an amendment because of its distinctive advantage of specific surface adsorption and resistance to hydrolysis [29]. In addition, limestone is also one of the oldest and broadest immobilizing agents [30], and the addition of lime plays an important role in increasing sediment pH and triggering the precipitation of metal oxides, carbonates, or hydroxides [31]. However, both kaolinite and limestone are rarely applied in dredged marine sediment.

Located in the northeast of China, Bohai Bay is a typical semi-enclosed bay, with shallow water and poor water exchange [32,33]. Adjacent to the Beijing–Tianjin city band and Bohai Rim economic circle, the coastal zones of Bohai Bay are one of the most densely urbanized and industrialized areas in China [34]. Recent research showed that quantities of metals were discharged directly into the Bohai Sea in 2018. In particular, 8237.88 kg of Pb was discharged [35]. Tianjin Port, located in Bohai Bay, is the largest comprehensive port in Northern China, where large-scale dredging activities are carried out along the coastal areas every year [34,36]. It was reported that a total of $466 \times 104$ m$^3$ of dredged materials were dumped into sites adjacent to the Tianjin Port between 2012 and 2013, resulting in a negative influence on the marine ecosystem in Bohai Bay; therefore, great importance should be attached to this issue [34,37,38].

In recent years, researches about the immobilization of metals in marine sediments have been carried out [39,40]. However, kaolinite and limestone were seldom used as immobilization agents to remediate metal pollution in marine sediment, and researches reflecting the influence on metals' mobility and transformation behaviors in the immobilization procedures were still rarely carried out, especially in the dredged marine sediment of Bohai Bay. Therefore, the primary objectives of this research are to evaluate the influence of pH, EC, salinity, TOC in marine sediment on the immobilization effect and to assess the effect of limestone and kaolinite, as amendments on immobilization of metals, on the basis of the sequential extraction procedure. This study is expected to provide a technical reference for the remediation of metal pollution in dredged marine sediments.

## 2. Materials and Methods

### 2.1. Collection and Pretreatment of Marine Sediment Samples

In June 2017, the marine surface sediment samples were collected using plastic shovels from coastal areas in Tianjin, Bohai bay. The sampling site was approximately 200 m from Bohai Bay beach (39°13' N, 117°58' E). The surface sediment samples (≤10 cm of depth) were collected and stored in PVC (polyvinyl chloride) tubes (60 cm). Then, the samples were taken back to the laboratory and immediately put into a freezer at 4 °C. After the large particles were removed, the sediment samples were air-dried naturally, crushed fully, and passed through a 2-mm sieve for bulk chemical analysis. A single homogeneous sediment sample was prepared by mixing individual samples.

## 2.2. Analytical Methods

The physicochemical properties of the sediment samples are shown in Table 1. Sediment pH was determined by a pH meter (Orion Star A211, Thermo Scientific, Tangerang, Indonesia) in ultrapure water using mass ratios of 1:2.5 (sediment to water). EC was measured in ultrapure water using mass ratios of 1:5 (sediment to water) by a portable multiparameter device (Orion Star A329, Thermo Scientific, Tangerang, Indonesia). TOC was measured with a TOC analyzer (Vario TOC Cube, Elementar, Langenselbold, Germany), and the grain size of the sediment was determined by a laser size analyzer (MalvernMastersizer 2000, Malvern Panalytical Ltd., Malvern, UK). Ultrapure water was used for all experiments to avoid contamination. The standard working solution, comprising three single-element standard solutions, was provided by the National Chemical Reagent Quality Inspection Center ($Zn(NO_3)_2$, $Pb(NO_3)_2$ and $Cu(NO_3)_2$, each with a concentration of 1 g/L). The basic physical–chemical properties of the sediments are presented in Table 4. The sediment background values of Zn, Pb, and Cu in Bohai Bay are 57 mg/kg, 11.5 mg/kg, and 19 mg/kg, respectively [41,42], and the concentration of the three metals are distinctly above the environment background values, particularly Zn, indicating the current severe pollution status of marine sediment in Bohai Bay.

**Table 1.** The physicochemical properties of the marine sediments. EC: electrical conductivity, TOC: total organic carbon.

| Property | | Sediment |
|---|---|---|
| pH | | 7.9 |
| EC (ms/cm) | | 8.08 |
| Salinity (‰) | | 4.50 |
| - | | - |
| TOC (%) | | 0.78 |
| Zn (mg/kg) | | 132.50 |
| Pb (mg/kg) | | 30.85 |
| Cu (mg/kg) | | 34.15 |
| | Sand | 4.47% |
| Grain size | Silt | 57.46% |
| | Clay | 38.07% |

## 2.3. Incubation Experiment

The homogeneous sediment sample was randomized into three groups (two immobilizing amendments plus one control). Three pots containing 40 g of sediment sample were used to conduct an incubation experiment. To study the immobilization effect of metals, a contaminant solution, artificially prepared with a concentration of heavy metals of 1 g/L, as suggested by Huang et al. [43], was added to the sediment sample. The synthetic solution containing metals was prepared by dissolving the solids ($Zn(NO_3)_2 \cdot 6H_2O$, $Pb(NO_3)_2$ and $Cu(NO_3)_2$) into the designed concentration. Moreover, to all groups, 8 mL of the synthetic solution, prepared as shown above, was artificially added. Then, the first one was used as a control group, and two pots were added to the other, along with the 2 g of kaolinite and limestone (which could be passed through 200-mesh sieve). The properties of the immobilization amendments are listed in Table 2. All of the compounds in the experimental pots were mechanically mixed for enough time with the stirring equipment, before homogeneous blending [44]. Next, the mixture of sediment with the additional solution and amendments was kept in the dark at room temperature (25 °C) for incubation to ensure a better immobilization effect with different periods. In previous studies on the immobilization of metals, the different incubation periods, from a week to three months [45–47], were applied. In this study, three time periods, including 1 day, 25 days, and 40 days, were set, and the sequential extraction experiment was conducted in each period to determine each fraction of Zn, Pb, and Cu.

**Table 2.** The properties of the immobilization amendments.

| Immobilization Agents | Molecular Formula | pH | EC (ms/cm) | Salinity (‰) |
|---|---|---|---|---|
| kaolinite | $Al_2Si_2O_5(OH)_4$ | 6.7 | 0.07 | 0.09 |
| limestone | $CaCO_3$ | 9.2 | 0.14 | 0.12 |

*2.4. BCR Sequential Extraction Procedure*

A modified BCR sequential extraction procedure of metals was applied. Each step of the modified BCR was briefly described in Table 3. The method divided the metals into four fractions: acid exchangeable fraction (F1) (F1 is an exchangeable and easily soluble fraction), reducible fraction (F2), oxidizable fraction (F3), and residual fraction (F4). Trace metals of Zn, Pb, and Cu in each fraction were analyzed by a volt-ampere spectrometer (797 VA Computrace, Metrohm, Herisaucity, Switzerland).

**Table 3.** The modified BCR sequential extraction procedure.

| Step | Fraction | Reagent | Procedure |
|---|---|---|---|
| 1 | acid exchangeable fraction | 40 mL of 0.11 mol/L CH3COOH | 22 ± 5 °C for 16 h, 3000 rpm for 20 min |
| 2 | reducible fraction | 40 mL of 0.5 mol/L $NH_2OH \cdot HCl$ | 22 ± 5 °C for 16 h, 3000 rpm for 20 min |
| 3 | oxidizable fraction | 10 mL of 8.8 mol/L $H_2O_2$, 50 mL of 1.0 mol/L $CH_3COONH_4$ | 1 h at 25 °C, 3 h at 85 ± 2 °C, twice; 22 ± 5°C for 16 h, 3000 rpm for 20 min |
| 4 | residual fraction | 0.1000 g remaining, 3 mL HCl, 2 mL $HNO_3$, 1 mL of $HClO_4$ and 3 mL of HF | 2 h at 110 °C, overnight, 2 h at 130 °C, increase to 150 °C until smoke gone, diluted to 10 mL |

*2.5. The Method Used to Evaluate the Immobilization Effect*

The mobility factor (MF%) has been defined as the percentage of metals in the acid exchangeable fraction in the cumulative total extracted amount of the metal, which was used to express the immobilization effect of different immobilizing agents [48,49]. Among the four fractions, which were extracted in sequential extraction, the first fraction (acid exchangeable fraction) contains the metals with the highest mobility and is readily available to organisms, thus having the most severe toxicity and ecological risk [50]. The small MF% value means the low mobility and bioavailability of metals, and thus a slight toxicity to marine ecosystems, suggesting a satisfactory immobilization effect. The mobility factor (MF%) could be calculated using the following Equation (1):

$$MF\% = \frac{F1}{F1 + F2 + F3 + F4} \times 100\% \tag{1}$$

*2.6. Quality Control*

The standard lake sediment, BCR 701 (European Commission, Joint Research Centre), was used to check the accuracy of the fraction analysis. A comparison of the concentration values of the first three fractions of BCR 701, determined in our laboratory, and that of certified values are shown in Table 4. The results demonstrated a good consistency with the reference values. In addition, the recovery of the sequential extraction procedure was calculated as follows in Equation (2):

$$\text{Recovery}(\%) = [ (C_{F1} + C_{F2} + C_{F3} + C_{F4}) / C_{\text{total concentration}} ] \times 100\% \tag{2}$$

where C represented the concentration of metals in the sediments [51]. The recovery rate ranged from 93% to 108%, suggesting the high reliability of the metals' fraction data obtained in this study. On the other hand, all the extraction procedures and the determination of concentration of metals included two replicates to guarantee the precision of the final results.

**Table 4.** Comparison of the results of our laboratory and certified values on BCR 701.

| Fraction | | Zn (mg/kg) | Pb (mg/kg) | Cu (mg/kg) |
|---|---|---|---|---|
| F1 | certified | 205 | 3.18 | 49.3 |
| | measured | 208.81 | 3.08 | 53.13 |
| F2 | certified | 114 | 126 | 124 |
| | measured | 113.45 | 118.63 | 122.36 |
| F3 | certified | 46 | 9.3 | 55 |
| | measured | 51.55 | 8.70 | 50.23 |

## 3. Results

### 3.1. The Properties of the Marine Sediments

The mobility and bioavailability of metals depend largely on pH, EC, organic adsorption, and the ionic coprecipitation process; therefore, these crucial parameters have the potential to change the dominant metal fractions in sediment [52]. The variation of these properties (pH, EC, salinity, and TOC) in the sediment samples, with different amendments over the incubation experiment time, is shown in Figure 1.

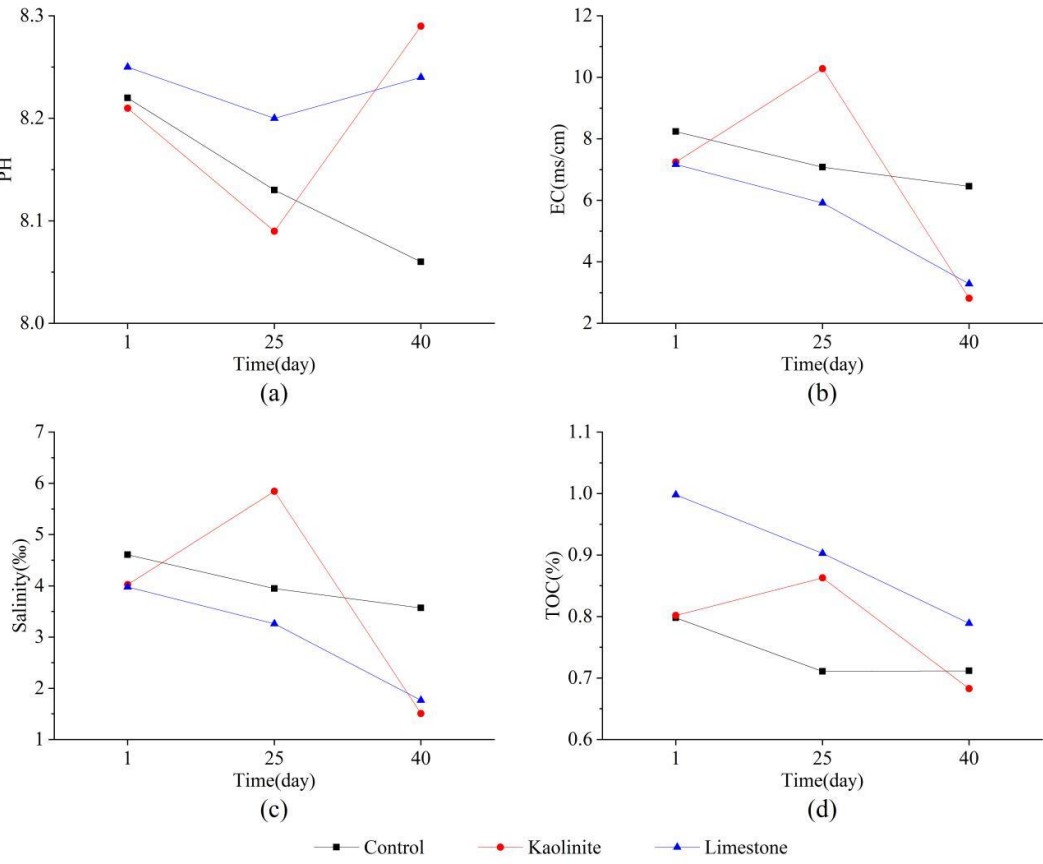

**Figure 1.** The properties of the sediment samples. (**a**) The pH values of the sediments in control, kaolinite and limestone groups with incubation time periods of 1 day, 25 days, and 40 days; (**b**) The EC values of the sediments in control, kaolinite and limestone groups with incubation time periods of 1 day, 25 days, and 40 days; (**c**) The salinity values of the sediments in control, kaolinite and limestone groups with incubation time periods of 1 day, 25 days, and 40 days; (**d**) The TOC values of the sediments in control, kaolinite and limestone groups with incubation time periods of 1 day, 25 days, and 40 days.

The pH value of the sediments in the control group decreases slightly, from 8.22 to 8.13 and then to 8.06 after 1 day, 25 days, and 40 days, respectively, from the beginning of the incubation experiment. After the addition of kaolinite and limestone, the sediment pH varies from 8.21 to 8.09 and then to 8.29, and from 8.25 to 8.20 and then to 8.24 at the same three periods as those shown above, respectively. It is worth noting that the addition of a high pH limestone (9.20) into the sediment sample raises the sediment pH slightly, compared with the control, in each experimental period.

The EC value in the control sediment sample shows a downward trend, from 8.238 ms/cm to 7.080 ms/cm and then to 6.459 ms/cm, after the incubation experiment periods of 1 day, 25 days, and 40 days, respectively. A similar variation trend is also observed in the sediment with a limestone amendment, the EC values of which are 7.170 ms/cm, 5.919 ms/cm, and 3.284 ms/cm, respectively. In particular, the EC values of the sediment sample with limestone are consistently lower than those of the control in each experimental incubation period. By contrast, the EC values of sediments with the kaolinite amendment decreased from 7.246 ms/cm on the first day to 2.815 ms/cm on the last day during the incubation experiment, with a sudden peak of 10.280 ms/cm on the 25th day.

The salinity values in the control sediments drop slightly from 4.609‰ to 3.947‰ and then to 3.570‰ after 1 day, 25 days, and 40 days, respectively, from the beginning of the incubation experiment. The salinity in the sediment with limestone group decreases from 3.980‰ to 3.261‰ and then to 1.766‰ at the same time periods shown above, respectively, which are moderately lower than those of the control group in each period. After the addition of kaolinite, the salinity is 4.024‰ on the first day of the incubation experiment, reaches 5.845‰ at the second period, and falls to its minimum of 1.508‰ at the end of the incubation experiment.

The TOC values of the sediment sample are 0.798%, 0.711%, and 0.712% in the control group after the incubation experiment periods of 1 day, 25 days, and 40 days, respectively. After kaolinite and limestone are added, the TOC values of both groups show an analogous decline. The kaolinite group decreases from 0.802% to 0.863% and then to 0.683%, and the limestone group decreases from 0.998% to 0.903% and then to 0.789% after 1 day, 25 days, and 40 days, respectively, from the beginning of the incubation experiment.

The properties of the sediment samples (pH, EC, salinity, and TOC) in the sediment samples with different immobilization amendments, including the control, kaolinite, and limestone groups, over three incubation time periods.

### 3.2. The Chemical Fractions of Metals in the Marine Sediments

The sequential extraction procedure was performed to characterize the contribution of each fraction in three metals, before and after the addition of amendments. The percentage of each fraction of the sum of four fractions is presented in Figure 2. The mobility and bioavailability of a trace metal are largely dependent on its distribution within geochemical fractions. Generally, it increases as follows: F1 > F2 > F3 > F4 [52]. Among all three groups, the residual fraction is dominant in Zn, with a proportion of nearly 60%, while F1, F2, and F3 account for only about 15%, 15%, and 10%, respectively. The results are in accordance with a previous study [53], revealing that the main proportion of Zn is bounded in the crystals of minerals, and its ecological risk was relatively low. The same main residual fraction is found for Pb; the average proportions of F1, F2, F3, and F4 were about 10%, 35%, 5%, and 50%, respectively. Pb was observed to be largely associated with F2, which was in agreement with the results of Marquez et al., and the combined binding of Pb to Fe and Mn oxides may be partly explained by a higher near-neutral pH [53]. The percentage of each fraction of Cu is ranked as follows F4 > F2 > F1 > F3, with the proportion of each at around 50%, 30%, 15%, and 5%, respectively. The dominant F4 proportion of Cu suggested its relative ecological risk, similar to Zn. In addition, Cu is strongly associated with F1 and F2, indicating its pH-dependent mobility. The result is in line with studies that rare earth elements sorption largely depends on pH, as well as hydrous manganese and ferric oxide content [54].

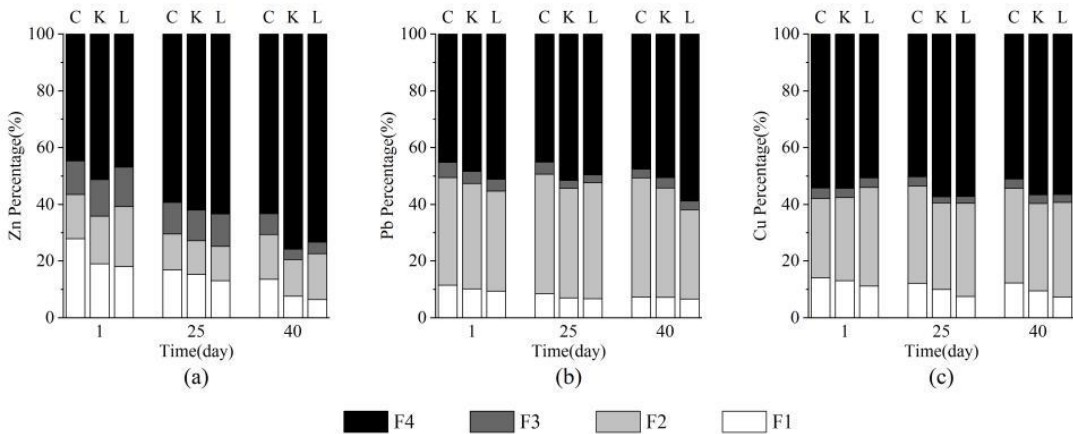

**Figure 2.** The distribution of the chemical fractions of metals in the sediments. C: control, K: kaolinite, and L: limestone. (**a**) The chemical fractions of Zn in control, kaolinite and limestone groups with incubation time periods of 1 day, 25 days, and 40 days; (**b**) The chemical fractions of Pb in control, kaolinite and limestone groups with incubation time periods of 1 day, 25 days, and 40 days; (**c**) The chemical fractions of Cu in control, kaolinite and limestone groups with incubation time periods of 1 day, 25 days, and 40 days.

After the immobilizing agents are added, each fraction varies according to the different extents. It can be noticed that F1 in Zn Pb and Cu is significantly reduced, while F4 in the three metals moderately increased with both amendments, compared with the control, indicating that the direct mobility and toxicity of the metals were alleviated. This is in line with other findings that metal bioavailability was inhibited with the addition of amendments [9,31]. In the kaolinite group, the F3 proportion of Zn, Pb, and Cu falls by 9.10%, 0.47%, and 0.21%, respectively, after 40 days of the incubation experiment, while that of F4 rises by 24.55%, 2.14%, and 2.28%, respectively. The F2 percentage of Zn decreases from 16.85% to 11.90% and then to 12.82% over the three incubation time periods, while no obvious variation was observed in that of Pb and Cu. In the case of the limestone group, both the F2 and F3 percentages of Zn, Pb, and Cu show a decreasing trend after 1 day, 25 days, and 40 days, from the beginning of the incubation experiment. The former drops by 5.07%, 3.83%, and 1.37%, respectively, and the latter drops by 9.75%, 1.05%, and 0.57% respectively.

Meanwhile, metals can be divided into three different types, based on their mobility and bioavailability, including their bioavailable fraction (F1), potentially bioavailable fraction (F2 and F3), and nonbioavailable fraction (F4) [54]. After comparing the variation of the chemical fractions, it is observed that the bioavailable fractions in the group of kaolinite and limestone are lower than those of the control for all three metals, while the potentially bioavailable and nonbioavailable fractions are obviously higher than those of the control for all three metals. It can be concluded that the mobile and bioavailable fractions of metals are transformed into comparatively stable fractions after the addition of immobilization agents.

## 4. Discussion

### 4.1. The Properties Influencing the Immobilization Effect

Previous research studies suggested that increasing sediment pH causes a weak competition of $H^+$ with metal ions for ligands (e.g., $OH^-$, $CO_3^{2-}$, $SO_4^{2-}$, $Cl^-$, and $S^{2-}$), and makes it easier for metal ions and ligands to combine into a relatively more stable form [50]. Cu has a great affinity for oxyhydroxides, and the immobilization of Cu in sediment is strongly pH-dependent [24,55]. In addition, the affinity of carbonates for Cu is a common reaction, causing a Cu deficiency in sediments in the presence of free $CaCO_3$ [56]; thus, the presence of limestone favors the combination of Cu and carbonates [57]. The same behaviors are also found in Zn and Pb, which can reduce the mobility of both metals in

sediments [58]. In the case of kaolinite, the hydroxyl groups ($\equiv$SOH) adsorbed in the surface or edge of kaolinite are amphoteric, and the surface charges are greatly sensitive to pH. At a high pH, $H^+$ is easily released into the solution from the surface hydroxyl groups ($\equiv$SOH $\rightarrow$ $\equiv$SO$^-$ + $H^+$), therefore leading to a combination of negatively charged $\equiv$SO$^-$ sites and metal species and forming metallic surface complexes, such as $\equiv$SOMe+ or $\equiv$SOMeOH [59].

The EC can act as a measure of the concentration of salts. [60]. The EC value in each experimental period is moderately lower than that in the control group, suggesting a lower soluble salt concentration [61]. A similar behavior was observed in previous researches on other amendments, probably because adding kaolinite and limestone into the sediment effectively slowed down the accumulation of salts [62,63]. A similar variation trend was observed in the salinity, as the EC of the sediments could be confirmed by a strong correlation between the two values [64]. As a previous study has pointed out, the presence of salts in sediment is strongly associated with complex ionic exchange and osmotic effects, which may contribute to the reduction of the mobility of metals in marine sediments [65].

The decline of the TOC value means that the organic carbon of sediment could be decomposable because of the addition of immobilization agents, which is consistent with the previous findings [66]. The immobilizing amendments decrease metal leaching by adsorption processes, which favor the formation of stable complexes with organic ligands [24]. It was reported that Cu easily forms complexes with organic matter, owing to the high stability of Cu compounds [67,68]. However, organic matter cannot be the dominant control over metal behavior when its concentration is relatively low [69]. Cation exchange and complexation with organic ligands were reported to be the primary Zn mobility-controlling mechanisms, while Al, Mn, and Fe oxides were less important [58]. The organic matter makes the formation of stable organometallic complexes easier, which could diminish the mobility of the metal ions in sediment [70].

*4.2. Immobilization Effect of Metals in Marine Sediments*

The mobility factor (MF%) value of Zn, Pb and Cu with different amendments is shown in Table 5. For the sediment sample without immobilizing agents, the MF% values were 27.85%, 16.88%, and 13.56% for Zn, 11.52%, 8.54%, and 7.33% for Pb, and 14.14%, 12.19%, and 12.30% for Cu, after the incubation experiment periods of 1 day, 25 days, and 40 days, respectively. All of the values show a decreasing trend, indicating some degree of a process of self-purification within the sediments [71]. In the case of the kaolinite group, the MF% values of Zn, Pb, and Cu were 19.10%, 10.18%, and 13.11%, respectively, on the first day of the incubation experiment, and declining to 7.68%, 7.25%, and 9.56%, respectively, on the 40th day, after experiencing a downward trend. After limestone was added, there was also a decline in the MF% value of Zn, Pb, and Cu, from 18.10%, 9.41%, and 11.17% to 6.48%, 6.67%, and 7.32%, respectively, over the incubation experiment time. The best immobilization effect was found in Zn, for which F1 drops by 11.42% and 11.61% in the kaolinite and limestone groups, respectively, and F4 is significantly increased by 24.55% and 26.43%, respectively. The MF% of kaolinite and limestone is obviously smaller than that of the control at each incubation time period, suggesting a satisfactory immobilization effect. Consequently, the addition of kaolinite and limestone can effectively reduce the bioavailability and toxicity of metals. It can be further observed that the MF% of Zn in the limestone group is moderately lower than that in the kaolinite group, and that for Pb and Cu is slightly lower than that in the kaolinite group, indicating that limestone generally has a better immobilization effect than kaolinite.

**Table 5.** The mobility factor (MF%) value of three metals with different amendments.

| Metals | Incubation Days | Control | Kaolinite | Limestone |
|---|---|---|---|---|
| | 1 | 27.85% | 19.10% | 18.10% |
| Zn | 25 | 16.88% | 15.32% | 13.05% |
| | 40 | 13.56% | 7.68% | 6.48% |
| | 1 | 11.52% | 10.18% | 9.41% |
| Pb | 25 | 8.54% | 7.09% | 6.71% |
| | 40 | 7.33% | 7.25% | 6.67% |
| | 1 | 14.14% | 13.11% | 11.17% |
| Cu | 25 | 12.19% | 10.07% | 7.54% |
| | 40 | 12.30% | 9.56% | 7.32% |

## 5. Conclusions

The determination of the properties of the sediments (pH, EC, salinity, and TOC) revealed that no obvious variations of pH were observed, while the EC, salinity, and TOC dropped moderately over the incubation experiment time because of the addition of the two amendments.

The results of the sequential extraction procedures showed that the mobile and bioavailable fractions of metals were transformed into relatively stable fractions with the addition of kaolinite and limestone. The calculation of MF% indicated that the value of Zn, Pb, and Cu decreased by 11.42%, 2.93%, and 3.55%, respectively, with kaolinite at the end of the incubation experiment, and it was 11.61%, 2.73%, and 3.85% in the limestone group, respectively. Thus, both limestone and kaolinite can be useful in immobilizing metals. The smallest MF% and largest reduction in metal mobility was found for limestone, suggesting that it has a better effect in immobilizing metals, compared with kaolinite.

**Author Contributions:** Methodology, software, formal analysis, investigation, data curation, writing—original draft preparation, W.Z.; writing—review and editing, X.W. and B.L.

**Funding:** This work was funded by the National Natural Science Foundation of China (NSFC), grant number 41106108.

**Conflicts of Interest:** The authors declare no conflict of interest. The funders had no role in the design of the study; in the collection, analyses, or interpretation of data; in the writing of the manuscript, or in the decision to publish the results.

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
