# Peer review of "An Ex-Situ Immobilization Experiment with Zn, Pb, and Cu in Dredged Marine Sediments from Bohai Bay, China"

_jmse, doi:10.3390/jmse7110394_

Round 1

Reviewer 1 Report

It is a nice study on Cu,Zn and Pb immobilization from marine sediments.

I have a few comments to improve the paper.

1) Do not use the term heavy metals anymore, check the papers for discussion on that and use metals, trace elements, toxic elements... or just uses Zn, Cu and Pb.

Pourret, O. and J.-C. Bollinger (2018). "“Heavy metal” - What to do now: To use or not to use?" Science of The Total Environment 610-611: 419-420.

Pourret, O. (2018). "On the Necessity of Banning the Term “Heavy Metal” from the Scientific Literature." Sustainability 10(8): 2879.

2) Regarding XRD, it seems that there is no difference, i am wondering why for kaolinite sample, intensity of Kln is so small.

I better call that "Powder pattern of samples", does the intensity have an arbitrary unit?

3) I guess more metal are coming from the North China Plain, in a recent study of one of my student, we show that metals are driven by iron and manganese oxides. Please consider that:

Liu, H., et al. (2018). "Impact of Hydrous Manganese and Ferric Oxides on the Behavior of Aqueous Rare Earth Elements (REE): Evidence from a Modeling Approach and Implication for the Sink of REE." International Journal of Environmental Research and Public Health 15(12): 2837.

4) Eventually authors should expand their discussion regarding fraction distribution of metals. It rather well known that at alkali pH limestone would be better than kaolinite, you can check methods in those 2 papers

Marquez, J. E., et al. (2018). "Effect of Cadmium, Copper and Lead on the Growth of Rice in the Coal Mining Region of Quang Ninh, Cam-Pha (Vietnam)." Sustainability 10(6): 1758.

Muyumba, D. K., et al. (2019). "Mobility of copper and cobalt in metalliferous ecosystems: Results of a lysimeter study in the Lubumbashi Region (Democratic Republic of Congo)." Journal of Geochemical Exploration 196: 208-218.
Author Response

Dear  Reviewer:

Thanks very much for your  comments concerning our manuscript entitled “The ex-situ immobilization experiment of heavy metals in dredged marine sediments from Bohai Bay, China” (jmse-598331). Those comments are all valuable and very helpful for revising and improving our paper, as well as the important guiding significance to our researches. We have studied comments carefully and  made correction which we hope meet with approval. Revised portion are marked in red in the paper.

We think the most questionable part in our manuscript is XRD experiment for analysis of clay minerals. We are very sorry for our negligence and we have made corrections. We originally planed to study the transformation of clay minerals to reveal their relation with behaviors of metals, but we must admit that XRD was not enough to explain the transformation of clay minerals.  In addition, we want to explain that the data we obtained is relative percentage of each kind of clay mineral not absolute percentage, and  it was a semi-quantitative analysis of clay minerals. We still tried to answer the questions that you raised, and we added the pre-treatment of XRD. After discussion with my co-authors, we decided to remove this part in our manuscript and conduct more experiments in our next paper in which fly ash will be used as immobilization amendment.

We have sent our manuscript to MDPI English editing system for extensive English checking and improvement of style. We tried our best to improve the manuscript and made some changes in the manuscript. These changes will not influence the content and framework of our paper. And here we did not list the changes but marked in red in revised paper and enclosed letter. Please see the attachment.

We appreciate for your warm work earnestly, and hope that the correction will meet with approval. Once again, thank you very much for your comments and suggestions.

Wensi Zhang

Reviewer 2 Report

This work, although not original in the idea presented, has a very important objective, by trying to study and test processes of immobilization of toxic metals in contaminated marine sediments after their dredging and before their disposal in appropriate places. However, it has some problems with the methodology used, the lack of explanation of many points, such as the sampling and refers to methods that were not necessary for anything, such as CEC. The main problem for me is how did they identify the clay minerals. I honestly do not believe in the identification of montmorillonite only by XRD without glycolation or saturation with cations and I do not understand how they did the quantification of these minerals. In this way, part of their discussion of the transformation of minerals after the amendments, no longer makes sense.

Another problem is the English writing which is not correct and all the text should be reviewed by someone who speaks and writes correctly in English. It is necessary to review the verb tenses of the sentences, the concordance between subject and verb and the punctuation.

The data discussion is the better part of this paper. It has a good chemical foundation for the explanation of metals immobilization. On the other hand, the mineralogical transformation that they say occurs after the addition of amendments is debatable.

Pag1:

Line 12 – No need the word “respectively”

Line 30 and 32 – Replace “were” by “is” and “was” by “is”

Line 42 – replace “Of various..” by “Among various…”

Pag2:

Lines55-56 – Not agree! Kaolinite has not a high exchange capacity. On the contrary! Among clay minerals is the group with the lowest CEC

Line 56-61 – Why not use these sentences in the present??

Line 77 – primary objectives of this research is…

Line 83 – dredged marine sediments.

Line 85 – It is better to write “Collection and pretreatment of marine sediments samples

Line 86 – What was the length of the tubes and the equipment used to collect the sediments?

Pag3:

Line 98 - What kind of treatment did the samples undergo before they were analyzed in the DRX? On slides? Aggregates? in normal? heated? glycol samples? Please specify.

Line 99 – CEC was measured by standard method (NY/T 295-1995). Please Specify further on what this method is based. However…. Why do you refer CEC measurement if you never refer these analyses and data throughout the text??

Line 100 – Is it really necessary?!? “reagents were purchased from Beijing culture and commerce center”

Line 102-103 - what are these solutions?

Line 104 – Table 4?? Will not be table 1??

Line 104-107 - In this experiment how did you obtain the values of the background sediments? In the same sites? And in 2011 sediments were not contaminated yet?? And how were the values of Zn, Cu and Pb of the composite sediment used in the experiment obtained? You have to refer it.

Table 1 – Pay attention to the precision of numerical results. Too much decimal places.

Line 111 – remove “respectively”. Not necessary!

Line 113 – …a contaminant solution artificially prepared with a concentration of heavy metals of 1g/L, as suggested by Huang et al. (43) was added to the sediment sample

Line 112 – Remove “ … in this study”

Line 116 – To all groups it was artificially added 8 mL of the sysnthetic solution prepared as above.

Line 117 – Review the English of this sentence “… and the left two pots were added with 2 g…”

Line 118 – Please state whether kaolinite and limestone are ground and to what size of grain

Pag4:

Line 132 – “Exchangeable fraction fraction” – remove one fraction

F1 is a exchangeable and easily soluble fraction.

Line 145 – Formula needs better image quality

Pag5:

Line 153 - Formula needs better image quality

Pag6:

Line 199-208: How was it possible to identify montmorillonite without glycolation of the sample ??? how was it possible to separate Chlorite from montmorillonite?? And how did you do the quantification of the 4 clay minerals groups???

Pag7:

Line 217: “.. was bounded in crystals of mineral…” Please review this sentnce.

Line 222: “it can be noticed that F1 of Zn, Pb and Cu all significantly reduced, while F4…”

Pag 8:

Line 229: “..no obvious variation was observed in Pb and Cu”

Line 235-238 – Rewrite this sentence “After trhe overall comparison … than that of control”.

Line 256: “ H+ is easily released into solution…”

Line 260: “The EC can act as a measure of nutrients for both cations and anions”. I don’t agree!! EC quantify how a given material conduct electricity and is a measure of the concentration of salts and metals in solution. Not nutrients!

Line 262: Replace “reserchers” by “researches”

Line 265: “between both values” or “between the two values”

Pag9:

Line 272: “..owing the high stability of…”

Line 273: “of it, is..”

Line 276: “The organic matter makes easier the formation of stable …”

Line 278-295: truthfully I do not believe in the identification of montmorillonite only by XRD without glycolation or saturation with cations, so I do not see any sense of this point

Author Response

Dear  Reviewer:

Thanks very much for your  comments concerning our manuscript entitled “The ex-situ immobilization experiment of heavy metals in dredged marine sediments from Bohai Bay, China” (jmse-598331). Those comments are all valuable and very helpful for revising and improving our paper, as well as the important guiding significance to our researches. We have studied comments carefully and  made correction which we hope meet with approval. Revised portion are marked in red in the paper.

We think the most questionable part in our manuscript is XRD experiment for analysis of clay minerals. We are very sorry for our negligence and we have made corrections. We originally planed to study the transformation of clay minerals to reveal their relation with behaviors of metals, but we must admit that XRD was not enough to explain the transformation of clay minerals.  In addition, we want to explain that the data we obtained is relative percentage of each kind of clay mineral not absolute percentage, and  it was a semi-quantitative analysis of clay minerals. We still tried to answer the questions that you raised, and we added the pre-treatment of XRD. After discussion with my co-authors, we decided to remove this part in our manuscript and conduct more experiments in our next paper in which fly ash will be used as immobilization amendments.

We have sent our manuscript to MDPI English editing system for extensive English checking and improvement of style. We tried our best to improve the manuscript and made some changes in the manuscript. These changes will not influence the content and framework of our paper. And here we did not list the changes but marked in red in revised paper and enclosed letter. Please see the attachment.

We appreciate for your warm work earnestly, and hope that the correction will meet with approval. Once again, thank you very much for your comments and suggestions.

Wensi Zhang
